# Polycyclic Aromatic Hydrocarbons Activate the Aryl Hydrocarbon Receptor and the Constitutive Androstane Receptor to Regulate Xenobiotic Metabolism in Human Liver Cells

**DOI:** 10.3390/ijms22010372

**Published:** 2020-12-31

**Authors:** Lisa Goedtke, Heike Sprenger, Ute Hofmann, Felix F. Schmidt, Helen S. Hammer, Ulrich M. Zanger, Oliver Poetz, Albrecht Seidel, Albert Braeuning, Stefanie Hessel-Pras

**Affiliations:** 1Department Food Safety, German Federal Institute for Risk Assessment (BfR), Max-Dohrn-Straße 8-10, 10589 Berlin, Germany; lisa.goedtke@bfr.bund.de (L.G.); heike.sprenger@bfr.bund.de (H.S.); albert.braeuning@bfr.bund.de (A.B.); 2Dr. Margarete Fischer-Bosch Institute of Clinical Pharmacology, Auerbachstr. 112, 70376 Stuttgart, and University of Tübingen, 72074 Tübingen, Germany; ute.hofmann@ikp-stuttgart.de (U.H.); uli.zanger@ikp-stuttgart.de (U.M.Z.); 3SIGNATOPE GmbH, Markwiesenstraße 55, 72770 Reutlingen, Germany; felix.schmidt@signatope.com (F.F.S.); hammer@signatope.com (H.S.H.); poetz@signatope.com (O.P.); 4Biochemical Institute for Environmental Carcinogens, Prof. Dr. Gernot Grimmer Foundation, Lurup 4, 22927 Grosshansdorf, Germany; albrecht.seidel@biu-grimmer.de

**Keywords:** polycyclic aromatic hydrocarbons, nuclear receptors, xenobiotic metabolism, liver, toxicity

## Abstract

Polycyclic aromatic hydrocarbons (PAHs) are environmental pollutants produced by incomplete combustion of organic matter. They induce their own metabolism by upregulating xenobiotic-metabolizing enzymes such as cytochrome P450 monooxygenase 1A1 (CYP1A1) by activating the aryl hydrocarbon receptor (AHR). However, previous studies showed that individual PAHs may also interact with the constitutive androstane receptor (CAR). Here, we studied ten PAHs, different in carcinogenicity classification, for their potential to activate AHR- and CAR-dependent luciferase reporter genes in human liver cells. The majority of investigated PAHs activated AHR, while non-carcinogenic PAHs tended to activate CAR. We further characterized gene expression, protein abundancies and activities of the AHR targets CYP1A1 and 1A2, and the CAR target CYP2B6 in human HepaRG hepatoma cells. Enzyme induction patterns strongly resembled the profiles obtained at the receptor level, with AHR-activating PAHs inducing CYP1A1/1A2 and CAR-activating PAHs inducing CYP2B6. In summary, this study provides evidence that beside well-known activation of AHR, some PAHs also activate CAR, followed by subsequent expression of respective target genes. Furthermore, we found that an increased PAH ring number is associated with AHR activation as well as the induction of DNA double-strand breaks, whereas smaller PAHs activated CAR but showed no DNA-damaging potential.

## 1. Introduction

Polycyclic aromatic hydrocarbons (PAHs) are ubiquitously occurring environmental pollutants that are formed during the incomplete combustion of organic compounds. Human exposure occurs predominantly via inhalation, ingestion, and dermal absorption [1]. The PAH family consists of more than 100 structurally different compounds with many of them exhibiting toxic, mutagenic, and/or carcinogenic properties [2]. Similar to the majority of genotoxins, most genotoxic PAHs are pro-carcinogens that need to be metabolically activated to exert their genotoxicity. PAHs often induce their own metabolism by binding to the aryl hydrocarbon receptor (AHR). After translocation into the nucleus and formation of a heterodimer with the AHR nuclear translocator (ARNT), this AHR/ARNT complex then binds to dioxin or xenobiotic response elements (DRE/XRE) in the promoter regions of its target genes, leading to the upregulation of several xenobiotic-metabolizing enzymes, such as cytochrome P450 monooxygenase 1A1 (CYP1A1) and 1A2 (CYP1A2) [3]. While both enzymes can detoxify PAHs to more hydrophilic compounds, they can also mediate the formation to reactive metabolic intermediates being electrophilical in nature, such as diol epoxides, which can readily form DNA adducts or bind to other macromolecules [4,5]. DNA adducts, especially in critical genes such as proto-oncogenes and tumor suppressor genes, may lead to mutations, therefore enhancing the risk to initiate cancer [6,7]. Of note, benzo[*a*]pyrene (BaP) is the only PAH classified as a group 1 carcinogen for being carcinogenic to humans according to the International Agency for Research on Cancer (IARC). In addition to its involvement in the regulation of different phase I and phase II xenobiotic-metabolizing enzymes, AHR is also critical for the development and maturation of many tissues via its role in cell cycle regulation [8,9] and for the control of adaptive immunity [10,11].

Beside the widely studied role of AHR and CYP1A1 in PAH (de)toxification [12,13], there are recent findings that some PAHs, such as fluoranthene (Flu), phenanthrene (Phe), and pyrene (Pyr), interact with the human and/or murine constitutive androstane receptor (CAR), with a subsequent induction of its prominent target gene CYP2B6 [14,15,16]. CAR (*NR1I3*) belongs to the nuclear receptor superfamily and is highly expressed in the liver. It plays a crucial role in the regulation of xenobiotic-metabolizing enzymes by upregulating several isoforms of the CYP2B, CYP2C, and CYP3A subfamilies [17,18,19]. CAR also controls diverse physiological and pathophysiological processes, such as energy metabolism, cell proliferation, and carcinogenesis [20]. CAR is activated by a broad spectrum of endogenous and exogenous substances, such as steroids and bile acids, but also phenobarbital and certain pesticidal active substances (reviewed by Molnár et al. [21]). Whether this activation has an impact on the carcinogenicity of PAHs remains completely unknown.

In this study, we performed a systematic analysis of the potential of different PAHs to interact with the human nuclear receptors AHR and CAR and the subsequent regulation of their respective target genes in human hepatoma cell lines. Nuclear receptor activation in HepG2 cells as well as target gene expression quantity and activity of CYP enzymes in HepaRG cells were measured after treatment with ten PAHs different in structure and IARC carcinogenicity classification (see Figure 1). Complementarily, we assessed the induction of DNA double-strand breaks (DSBs) via the detection of phosphorylated H2AX (γH2AX) as a toxicologically relevant endpoint reflecting the DNA-damaging potential of the test compounds.

## 2. Results

### 2.1. Cytotoxicity

Cytotoxicity was analyzed beforehand for all PAH concentrations using the 3-(4,5-dimethylthiazol-2-yl)-2,5-diphenyltetrazolium bromide (MTT) cell viability assay. Cell viability was determined 24 h or 48 h after treatment for HepG2 or HepaRG cells, respectively. None of the concentrations used in this study decreased cell viability of HepaRG or HepG2 cells below 80% (see Figure 2).

### 2.2. Activation of the Human Receptors AHR and CAR

Activation of AHR and CAR was assessed using dual luciferase reporter gene assays in HepG2 cells. In general, a concentration-dependent activation of AHR (Figure 3A) and CAR (Figure 4A) was observed for all PAHs. AHR activation was substantially induced by benzo[*k*]fluoranthene (BkF) > benzo[*b*]fluoranthene (BbF) > benzo[*j*]fluoranthene (BjF) > BaP > benzo[*a*]anthracene (BaA). CAR-dependent luciferase activity was significantly increased by BbF > BjF > Flu, and to a lesser extent also by BaP, benzo[*c*]phenanthrene (BcPhe), Phe, and Pyr. Interestingly, PAHs that were weak AHR agonists were more likely to interact with CAR and vice versa. Exceptions to this observation were BbF and BjF, which led to strong activation of both AHR and CAR.

### 2.3. Gene Expression Analysis

To verify the results from the receptor activation analysis, the expression of AHR target genes, namely *CYP1A1* and *CYP1A2* (Figure 3B), and of the CAR target gene *CYP2B6* (Figure 4B) was analyzed in the human hepatoma cell line HepaRG using quantitative real-time PCR (qRT-PCR). Generally, most PAHs investigated led to a concentration-dependent increase in gene expression. Expression patterns of *CYP1A1* and *CYP1A2* mostly resembled one another. *CYP1A1* and *CYP1A2* levels were upregulated by all PAHs with BkF > BbF > BjF > BaA ≈ BaP being most potent, which is in good agreement with the observations made from the receptor activation analysis. In contrast, BcPhe as well as non-carcinogenic Flu, Phe, and Pyr affected *CYP1A1* and *CYP1A2* levels only weakly. Interestingly, *CYP2B6* was substantially upregulated by the same PAHs, along with BbF and BjF.

### 2.4. CYP Quantification

Previous studies described that gene expression does not always correlate with the amount of protein [23,24,25]. To investigate whether the effects observed for receptor activation and gene expression can be confirmed at the protein level, the regulation of CYP enzyme amounts was examined as another important endpoint. Figure 3C displays the regulation of CYP1A1 and CYP1A2, and Figure 4C of CYP2B6, respectively. 

Abundances of CYP1A1 and CYP1A2 were below the lower limit of quantification (LLOQ) in the solvent control. For normalization purposes, the value half of the LLOQ was used for calculations [26]. In general, all PAHs led to a concentration-dependent increase in CYP enzyme abundances. Over the incubation times investigated, the initially strong upregulation of CYP1A1 lessened, while CYP1A2 abundance increased, respectively. Treatments with BaA, BaP, BbF, BjF, BkF, and DbP (dibenzo[*a,l*]pyrene) led to the most pronounced increase of CYP1A1 and CYP1A2 protein content. In contrast, especially BcPhe, Flu, Phe, and Pyr were found to upregulate CYP2B6 protein levels significantly. This is in line with the previous observation that these PAHs were more likely to interact with CAR, thus primarily upregulating typical CAR genes.

### 2.5. CYP Activity

Complementarily to the quantification of protein abundances, we also examined the regulation of CYP activity. After PAH treatment, cells were incubated with substrates specific for CYP enzymes of interest, and later the abundances of metabolites formed were quantified, standing proportional to CYP activity. Here, we focused on CYP1A2 (Figure 3D) and CYP2B6 (Figure 4D). For other CYP activities measured, see Appendix A “Underlying Data”.

Overall, a concentration-dependent increase in CYP activity was found. In line with the previous findings, BaA, BaP, BbF, BjF, and BkF also led to the most pronounced increase in CYP1A2 activity. While the activity of CYP1A2 was not substantially affected after treatment with BcPhe, Flu, Phe, and Pyr, we observed CYP2B6 activity to be strongly increased. In general, the CYP activity pattern showed a strong resemblance to the protein abundance pattern, as seen in Figure 4.

### 2.6. DNA Double-Strand Breaks (DSBs)

Phosphorylation of the histone variant H2AX represents a sensitive marker for DSBs occurring after exposure to genotoxic compounds [27,28]. To assess the DNA-damaging potential of the investigated PAHs, γH2AX formation was analyzed in HepaRG cells by normalizing γH2AX fluorescence intensity to the 4′,6-diamidino-2-phenylindole (DAPI) signal (cell number).

Figure 5 shows representative images of γH2AX (red) and DAPI (blue) staining, as well as a merged picture 48 h posttreatment with 10 µM of each PAH. Figure 6 depicts quantitative analyses of the γH2AX fluorescence intensities. γH2AX levels were strongly increased in a time-dependent manner after treatment with the positive control etoposide (Eto), a topoisomerase inhibitor [29]. Most PAHs led to a time- and concentration-dependent increase in γH2AX formation with 10 µM DbP leading by far to the highest γH2AX levels measured, followed by BaP and BbF. As expected for non-carcinogenic compounds, Flu, Phe, as well as Pyr did not induce H2AX phosphorylation substantially.

## 3. Discussion

PAHs represent a large group of widely distributed environmental toxicants derived from natural and anthropogenic sources. Some exert mutagenic, carcinogenic, or teratogenic properties [30]. However, systematic analyses on their potential to activate nuclear receptors and the subsequent induction of target genes are rare.

Thus, in the present study, we investigated ten PAHs for their potential to interact with AHR and CAR and analyzed for subsequent regulation of their respective target genes *CYP1A1* and *CYP1A2*, as well as *CYP2B6*. As a toxicologically relevant endpoint, we examined the formation of DSBs to investigate the correlation between receptor activation, enzyme induction, and genotoxic effects. The choice of compound concentrations was based on the lead substance BaP. With an oral intake of 250 g grilled meat (containing 157 µg BaP/kg [31]) in an estimated stomach volume of 500 mL, the calculated concentration translates to 0.3 µM BaP. Therefore, the lowest BaP concentration used in this study was close to that of possible human exposure. As the liver is a main organ for PAH metabolism, the human hepatoma cell line HepaRG was chosen as cell culture model in the present study. Once differentiated, they are considered to be the cell line most similar to human primary hepatocytes, forming a mixed population of biliary and hepatocyte-like cells expressing a great variety of xenobiotic-metabolizing enzymes, including CYP1A1, CYP1A2, and CYP2B6, as well as receptors such as AHR, CAR, and pregnane X receptor (PXR) [32,33].

It is well studied that various PAHs activate AHR [34,35]. As expected, we showed that most of the investigated PAHs substantially activated AHR. In detail, our observations regarding AHR activation are in line with previous studies showing stronger AHR agonistic activity of BkF and BbF than of BaP [36,37]. BcPhe, as well as non-carcinogenic Flu, Phe, and Pyr, did not show a considerable potential to activate AHR, which is in good agreement with the observation of Sjögren et al. [38] who found low AHR binding affinities for both Flu and Pyr. As representative AHR target genes, we chose to investigate CYP1A1 and/or CYP1A2 at the mRNA and protein abundance and activity level. Here, a strong positive correlation between AHR activation, especially by BaP, BbF, BjF, and BkF, and subsequent induction of CYP1A1 and CYP1A2 was observed. Interestingly, although being controlled by a bidirectional promoter [39], we found CYP1A1 and CYP1A2 protein levels to be differentially regulated, with CYP1A1 being rapidly and strongly induced after only 12 h, while CYP1A2 abundance as well as activity increased over time, reaching a maximum level after 24 h. In comparison to CYP1A2, basal hepatic expression levels of CYP1A1 were very low but can reach high levels following induction [40].

Because recent studies reported that the non-carcinogenic Flu, Phe, and Pyr are agonists of CAR with subsequent induction of *CYP2B6* gene expression [14,15,16], we also analyzed the PAHs investigated in this study for their potential to activate CAR. Here, we could verify the aforementioned findings. Furthermore, we also observed BbF, BcPhe, and BjF to activate CAR, which to the best of our knowledge has not been described before. Luckert et al. [41] demonstrated that BcPhe was able to induce the *CYP3A4* promoter via activation of PXR. This may be supportive to our observations as CAR and PXR partially share ligands and target genes [42,43]. Although BcPhe as well as non-carcinogenic Flu, Phe, and Pyr led to a less pronounced CAR activation as compared with BbF and BjF, they induced the expression of the CAR target gene *CYP2B6* to a greater extent at the mRNA and protein abundance and activity levels. Since PAHs are known to induce their own metabolism, it may be assumed that PAHs, which upregulate CYP2B6 via CAR activation, are also metabolized by CYP2B6. Unfortunately, information about CYP2B6-mediated PAH biotransformation is rare. Previously, it was reported that CYP2B6 metabolizes BaP to 3- and 9-phenols and to trans-dihydrodiols [31]. However, most studies suggest that the majority of PAHs, including those that induced the expression of CYP2B6 in our study, are predominantly metabolized by members of the CYP1 subfamily [44,45].

In order to evaluate the DNA damaging potential of PAHs in the present study, we assessed H2AX phosphorylation as a surrogate marker for DSBs [3]. γH2AX is formed rapidly at the site of damage and can be detected in positive cells by immunostaining. We found that DbP led to the highest γH2AX formation, followed by BaP, BbF, and BaA. Previous studies already reported DbP to be substantially more carcinogenic than BaP in animal models and possibly to be the most potent carcinogenic PAHs tested so far [46,47,48,49]. Our findings demonstrate that mainly AHR-activating PAHs led to a time-dependent increase in DSBs, suggesting that induction of CYP1A1 seems to promote PAH bioactivation and therefore mediate genotoxic events such as γH2AX formation. Numerous in vitro studies have not only demonstrated that AHR activation is a major toxic mode of action but also that CYP1A1 is involved in the bioactivation of BaP into reactive intermediates capable of binding to DNA and proteins. However, in vivo toxicity studies with CYP1A1-deficient mice suggest that inducible CYP1A1 is far more important in detoxification than in metabolic activation [12,50]. Whether these contrary findings are due to inter-species differences (e.g., human and rodent) or due to in vitro–in vivo differences remains unclear.

The underlying principle behind the unique interaction pattern observed for the examined PAHs with AHR and CAR, respectively, is unknown to us. However, PAH ring number might be a critical factor as CAR-activating PAHs consisted of three or four aromatic rings, whereas the AHR-activating PAHs evaluated in this study were mostly made of five rings. This might explain why PAHs of lower molecular weight, such as Flu or Pyr, are not considered to be carcinogenic [2] as they tend to activate the CAR/CYP2B6 axis that does not seem to be majorly involved in PAH bioactivation. However, CAR agonists, such as phenobarbital, often act as tumor promotors in vivo [51]. If this also applies to CAR agonistic PAHs, it needs further elucidation in the future. In contrast to the finding that small PAHs primarily activated CAR, PAHs with higher ring numbers were observed to activate mainly AHR/CYP1A and to cause DNA-damaging effects. This relation is supported by previous observations where PAHs with high ring numbers were linked to an elevated DNA-damaging potential: High molecular weight PAHs consisting of six rings (i.e., dibenzopyrenes isomers) displayed a high AHR binding affinity together with an even higher carcinogenic potential than BaP [37,52]. Furthermore, the probability that certain PAHs exert mutagenic and carcinogenic activity increases with higher ring numbers, as structural arrangements might lead to a bay or fjord region—a structural pocket that is associated with higher reactivity and carcinogenic potential [1].

In conclusion, our study suggests a certain specificity of PAHs with respect to receptor activation and subsequent enzyme induction on mRNA and protein abundance and activity levels. Exceptions were BbF and BjF, which were observed to be agonists for both AHR and CAR and subsequently induced CYP1A1/CYP1A2 as well as CYP2B6, respectively. Furthermore, the majority of the ten PAHs tested in this study, in particular those with carcinogenic potential, were observed to (a) activate AHR; (b) upregulate CYP1A1/CYP1A2 at the mRNA and protein abundance and activity levels; and (c) lead to the formation of DSBs. In contrast, non-carcinogenic PAHs primarily activated CAR and subsequently induced CYP2B6. Furthermore, they did not substantially introduce DSBs, pointing toward the hypothesis that toxicity by PAHs in HepaRG cells is mostly mediated through the AHR/CYP1A axis.

## 4. Materials and Methods

### 4.1. Chemicals

All PAHs were kindly provided by Dr. Albrecht Seidel (Biochemical Institute for Environmental Carcinogens, Prof. Dr. Gernot Grimmer Foundation, Großhansdorf, Germany), dissolved in DMSO with a final concentration of 10 mM and stored at −80 °C. Atorvastatin calcium salt and S-mephenytoin were purchased from Toronto Research Chemicals (North York, ON, Canada). Amodiaquine dihydrochloride dehydrate, bupropion hydrochloride, and tolbutamide were obtained from Supelco (Bellefonte, PA, USA). Phenacetin and propafenone hydrochloride were from Sigma-Aldrich (Taufkirchen, Germany). All other chemicals were purchased from Merck (Darmstadt, Germany) or Sigma-Aldrich (Taufkirchen, Germany) in the highest available purity and used as received.

### 4.2. Cell Culture, Seeding and Treatment

HepaRG cells (Biopredic HPR101, St. Gregoire, France) were cultured according to the established protocol in William’s E medium (PAN-Biotech GmbH, Aidenbach, Germany) supplemented with 10% (*v/v*) heat-inactivated fetal calf serum (FCS, PAA, Pasching, Austria), 100 U/mL penicillin and 100 µg/mL streptomycin (P/S), 5 µg/mL insulin (PAN-Biotech GmbH, Aidenbach, Germany), and 50 µM hydrocortisone hemisuccinate (HHS, Sigma Aldrich). For assessing γH2AX formation and cell viability, cells were seeded in 96-well plates at a density of 9000 cells/well. For gene expression analysis and CYP protein quantification, cells were seeded in 6-well plates at a density of 200,000 cells/well. For CYP activity measurements, cells were seeded in 24-well plates at a density of 55,000 cells/well. The following protocol applied for all plates: After seeding, HepaRG cells were grown for two weeks in the aforementioned proliferation medium. For another two weeks, the proliferation medium was supplemented with 1.7% dimethyl sulfoxide (DMSO). Cells were maintained in an incubator at 37 °C and 5% CO2 with a water reservoir for humidity control, and the medium was renewed every 2–3 days. Prior to treatment, cells were adapted for 2–3 days to the treatment medium, which was similar to the proliferation medium but with only 2% FCS and 0.5% DMSO. Test substances were diluted in the treatment medium as well (final DMSO concentration 0.5%).

The human hepatocellular carcinoma cell line HepG2 was obtained from the European Collection of Authenticated Cell Cultures (ECACC 85011430, Porton Down, UK). Cells were cultured in high glucose Dulbecco’s modified Eagle’s medium (DMEM, PAN-Biotech GmbH, Aidenbach, Germany) supplemented with 10% (*v/v*) heat-inactivated FCS (Capricorn Scientific GmbH, Ebsdorfergrund, Germany) and P/S (100 U/mL / 100 µg/mL, Capricorn Scientific GmbH, Ebsdorfergrund, Germany) at 37 °C and 5% CO2 with a water reservoir for humidity control. Cells were passaged every 2–4 days at a confluence of about 80–90%. For assessing dual luciferase reporter gene activity and cell viability, HepG2 cells were seeded in 96-well plates at a density of 20,000 cells/well and used for experiments 24 h post seeding.

### 4.3. Reporter Gene Activity via Dual Luciferase Assay

HepG2 cells were used for analysis of receptor/promoter activation because of their high transfection rate. A detailed description of the assay principle and plasmids has been made previously [53]. Dual luciferase assays were performed as recently described [14]. Briefly, 24 h post seeding, HepG2 cells were transiently transfected with the respective plasmids (see Appendix A “Plasmids and Primers”, Appendix A) using TransIT-LT1 (Mirus Bio LCC, Madison, WI, USA) in a relation of 3:1 (TransIT-LT1 (µL): amount of plasmids (µg)). For CAR transactivation, cells were transfected with a pcDNA5-GAL4/DBD-hCAR/LBD(+3aa) plasmid containing the human CAR ligand-binding domain fused to the DNA-binding domain of the transcription factor GAL4, and a pGAL4-(UAS)5-TK-LUC plasmid that encodes the firefly luciferase reporter gene under the control of a GAL4-specific upstream activation sequence [41]. Since HepG2 cells already contain a sufficient amount of endogenously occurring AHR, the cells were transfected for AHR activation with only one plasmid—namely, pGL4.26-XRE6, which contains an AHR-specific DNA response element with a downstream sequence coding for the firefly luciferase reporter gene. It was cloned using the digestion sites KpnI (5′-end) and HindIII (3′-end) and inserting six repeats of the XRE-DNA sequence 5’-GAGTTCTCACGCTAGCAGATT-3 into the vector pGL4.26 (Promega, Madison, WI, USA). Additionally, the plasmid pcDNA3-Rluc was co-transfected in both assays for normalization purposes, encoding the Renilla luciferase under the control of a cytomegalovirus promoter. After 6 h of transfection, cells were treated with 1, 5, or 10 µM of each PAH for 24 h. Five micromolar 3-methylcholanthrene (3-MC) and 10 µM 6-(4-Chlorophenyl)imidazo[2,1-b][1,3]thiazole-5-carbaldehyde O-(3,4-dichlorobenzyl)oxime (CITCO) served as positive controls for AHR and CAR activation, respectively. Fold changes and standard deviations can be found in the Appendix A “Underlying Data”. After cell lysis and centrifugation, luciferase activities were analyzed as described before [41,54]. 

### 4.4. RNA Isolation and Quantitative Real-Time PCR (qRT-PCR)

For gene expression analysis, HepaRG cells were seeded, differentiated, and adapted to the treatment medium as described above. Afterward, cells were incubated with 1, 5, or 10 µM of each PAH for 24 h. After treatment, cells were washed with ice-cold phosphate-buffered saline (PBS), and total RNA was extracted using the RNeasy Mini Kit (Qiagen, Hilden, Germany), following the instructions of the manufacturer. Quality and quantity of the isolated RNA were determined with Tecan’s NanoQuant Plate (Tecan group, Männedorf, Switzerland). One microgram of RNA was then reverse transcribed using the High Capacity cDNA Reverse Transcription Kit (Applied Biosystems, Foster City, CA, USA). According to the manufacturer’s protocol, thermal cycling conditions were chosen for two-step reverse transcriptase PCR. qRT-PCR was performed in 384-well plates on a 7900HT fast real-time PCR system (Thermo Fisher Scientific, Waltham, MA, USA) using Maxima SYBR Green/Rox qPCR Master Mix (Thermo Fisher Scientific, Waltham, MA, USA) with 300 µM of each primer (see Appendix A “Plasmids and Primers”, Appendix A) and 1 µL cDNA in a total volume of 10 µL. Thermal cycling comprised an initial denaturation step at 95 °C for 10 min, followed by 40 cycles of denaturation at 95 °C for 10 s, annealing at 60 °C for 15 s, and extension at 72 °C for 20 s, and eventually a dissociation stage at 95 °C, 60 °C, and 95 °C for 15 s, respectively. For the relative quantification of cDNA content according to the 2^−ΔΔCt^ method [55], Ct values were normalized to the geometric mean of β-actin (*ACTB*) and glyceraldehyde 3-phosphate dehydrogenase (*GAPDH*) as housekeeping genes and referred to solvent-treated cells (set to a relative gene expression value of 1.0).

### 4.5. CYP Quantification

CYP quantification was performed as described by Braeuning et al. [56] with minor modifications. Briefly, HepaRG cells were seeded, differentiated, and adapted to the treatment medium as described above. Cells were then treated with 1, 5, or 10 µM of each respective PAH (final DMSO concentration 0.5%) for 12, 24, or 48 h. Afterward, the medium was discarded, and cells were washed twice with ice-cold PBS. Five hundred microliters lysis buffer [56] was added to each well and incubated at 4 °C for 2 h under constant shaking. Cell lysates were transferred to a fresh reaction tube and snap frozen in liquid nitrogen. Each experimental series was repeated independently three times. Prior to analysis, the total protein amount was determined via bicinchoninic acid (BCA) assay (Thermo Scientific, Waltham, MA, USA), and 60 µg protein was proteolyzed using trypsin (Pierce Trypsin Protease, MS-grade; Thermo Scientific, Waltham, MA, USA). Surrogate peptides and internal standard peptides were precipitated using triple X proteomics (TXP) antibodies. Peptides were then eluted and quantified using a modified version of the previously described LC–MS methods [56]: There, 10 min and 20 min parallel reaction monitoring (PRM) methods are described. In the present project, four CYPs, which had been analyzed with a 20 min gradient before, were transferred to the 10 min method (UltiMate 3000 RSLCnano and PRM–QExactive Plus; Thermo Scientific, Waltham, MA, USA). Raw data were processed using Skyline software (MACOSS Lab, Department of Genome Sciences, University of Washington, Seattle, WA, USA) and TraceFinder 4.1 (Thermo Scientific, Waltham, MA, USA). Peptide amounts were calculated by forming the ratios of the integrated peaks of the endogenous peptides and the isotope-labeled standards. Quantities of proteins were reported normalized as fmol per µg extracted protein.

### 4.6. CYP Activity

Preparations for assessing CYP activity were done as described by Voss et al. [57] with minor modifications. In short, HepaRG cells were seeded in 24-well plates at a density of 55,000 cells/well, differentiated and adapted to the treatment medium as described above. Cells were then incubated with 1, 5, or 10 µM of each respective PAH for 12, 24, and 48 h. Afterward, cells were washed with PBS and incubated for 3 h at 37 °C with 500 µL of a mix of substrates for CYP1A2 (phenacetin, 50 µM in DMSO), CYP2B6 (bupropion, 25 µM in H2O), CYP2C8 (amodiaquine, 5 µM in H2O), CYP2C9 (tolbutamide, 100 µM in acetonitrile), CYP2C19 (mephenytoin, 100 µM in acetonitrile), CYP2D6 (propafenone, 5 µM in methanol), and CYP3A4 (atorvastatin, 35 µM in 50% acetonitrile) diluted in assay medium. To stop the reaction, the supernatant was transferred to fresh tubes containing formic acid (final concentration 50 mM). Metabolite formation was measured with a mass-spectrometric approach as previously described [58,59] and stood proportional to the level of active enzymes.

### 4.7. γH2AX Immunofluorescence Staining and Microscopy

Phosphorylation of the histone H2AX, then called γH2AX, was analyzed via immunofluorescence staining, as recently described [14]. Etoposide (Eto, 25 µM; Cayman Chemicals, Hamburg, Germany) served as a metabolism-independent positive control. HepaRG cells were seeded and preconditioned as described above. After treatment with 1, 5, or 10 µM of each PAH for 12, 24, or 48 h, the incubation medium was discarded. Then, cells were washed with PBS, fixed with 50 µL/well ice-cold methanol (100%) for 30 min at 4 °C, and blocked with 50 µL/well blocking solution (1% bovine serum albumin (BSA) in PBS-T (0.1% Tween-20 in PBS (*v/v*)) at room temperature under gentle shaking for 1 h. Blocking solution was removed and cells were incubated with 40 µL/well of primary anti-γH2AX antibody (anti-phospho-histone H2A.X (Ser139), clone JBW301, Merck, Darmstadt, Germany, 1:500 in blocking solution), followed by a secondary antibody (Alexa Fluor 647, F(ab’)2-Goat anti-Mouse IgG (H+L), Invitrogen, Fisher scientific, Newington, NH, USA, 1:400 in blocking solution) at room temperature under gentle shaking for 1 h in the dark, respectively. Cells were counterstained for 30 min with 4′,6-diamidino-2-phenylindole (DAPI, 50 µL/well, 3 µM in PBS). Unless described otherwise, cells were washed thrice with PBS-T after each step. Fluorescence intensity was measured (AF647: 653 Ex/668 Em, DAPI: 353 Ex/465 Em) using a Celldiscoverer 7 (Zeiss, Oberkochen, Germany) with a 5x objective and 1x tube lens. Microscopy images were captured using an Axiocam 506 (Zeiss, Oberkochen, Germany) and Zen Blue software (Zen 3.1 (blue edition), Zeiss, Oberkochen, Germany). The AF647 signal was normalized to the DAPI signal and referred to solvent control.

### 4.8. Statistical Analysis

All data are represented as heatmaps generated by the R-package ComplexHeatmap [60]. Generally, mean values were calculated from the individually measured raw data. Fold changes with regard to the respective solvent control were determined and then log2-transformed. The underlying fold changes and standard deviations can be found in the Appendix A “Underlying Data”. Statistical analysis was performed using Student’s *t*-test in R version 4.0.2 [61]. Statistical significance was assumed at *p* < 0.05.

## Figures and Tables

**Figure 1 ijms-22-00372-f001:**
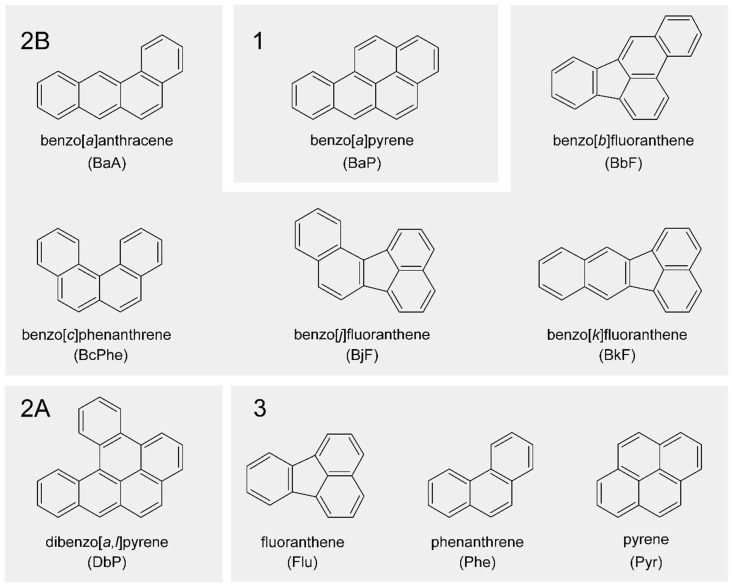
Chemical structures of polycyclic aromatic hydrocarbons (PAHs) used in this study in alphabetical order. PAHs are clustered additionally according to their International Agency for Research on Cancer (IARC) classification. Group (**1**): carcinogenic to humans; group (**2A**): probably carcinogenic to humans; group (**2B**): possibly carcinogenic to humans; group (**3**): not classifiable as to its carcinogenicity to humans [22].

**Figure 2 ijms-22-00372-f002:**
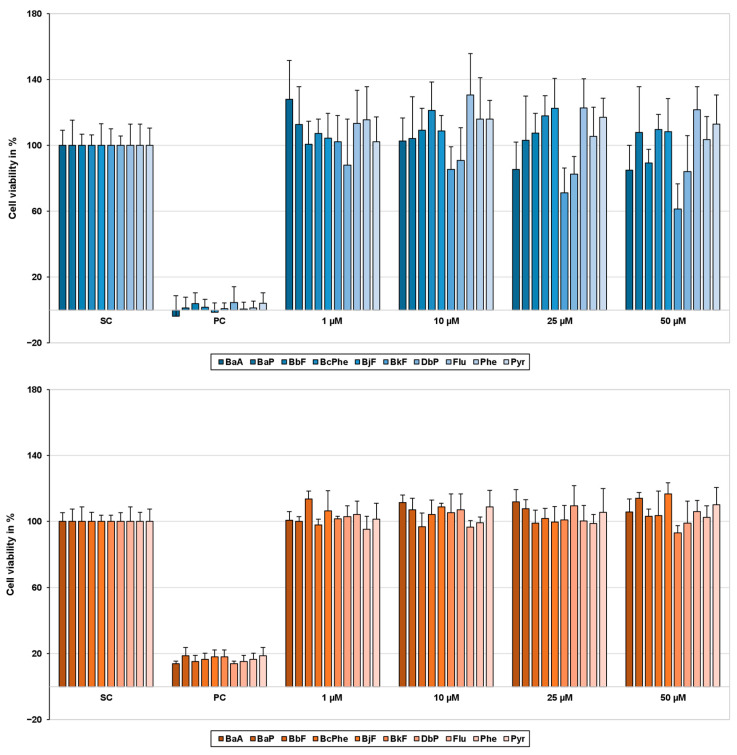
Cell viability of HepG2 (blue) and HepaRG cells (orange) 24 h and 48 h after polycyclic aromatic hydrocarbon (PAH) treatment, respectively. To determine cell viability, the 3-(4,5-dimethylthiazol-2-yl)-2,5-diphenyltetrazolium bromide (MTT) assay was performed. After treatment with the test compounds, 10 µL MTT (5 mg/mL in PBS) was added to each well and incubated for 2 h (37 °C, 5% CO_2_). To dissolve the blue-colored formazan crystals, incubation medium was exchanged for 130 µL/well pre-warmed desorption solution (0.7% sodium dodecyl sulphate (SDS) in propan-2-ol) and gently shaken for 15 min. Afterward, absorbance was measured at 590 nm on a Tecan Infinite M200 Pro spectrophotometer (Tecan group, Männedorf, Switzerland). Obtained data were normalized to the solvent control (SC); 0.01% Triton-X 100 served as a positive control (PC). Results are expressed as mean + SD, *n* = 3.

**Figure 3 ijms-22-00372-f003:**
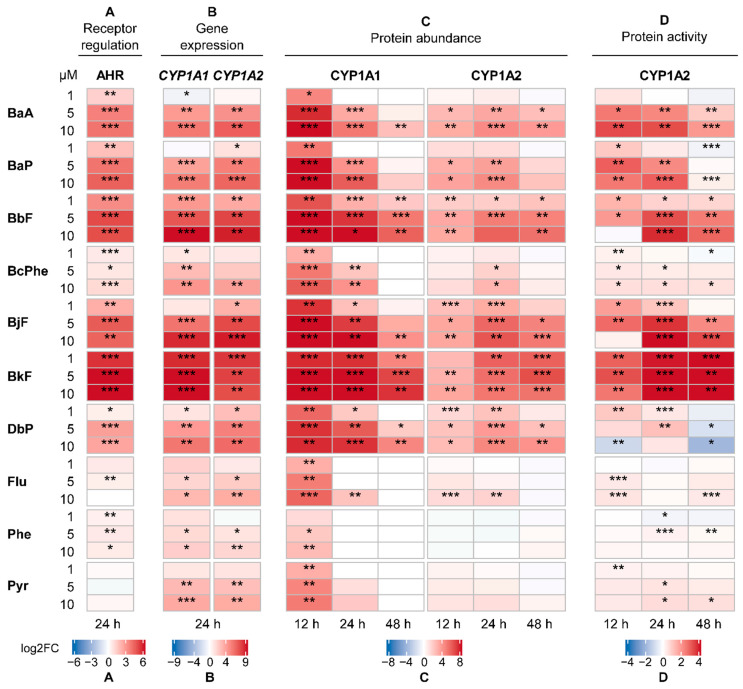
Cumulated heatmaps of receptor regulation (**A**), gene expression (**B**), protein abundance (**C**), and protein activity (**D**) of aryl hydrocarbon receptor (AHR) and its targets cytochrome P450 monooxygenase 1A1 (CYP1A1) and 1A2 (CYP1A2), respectively. Respective experiments were conducted as described in the Materials and Methods section. For protein abundance and activity, the following applied: For values below the respective lower limit of quantification (LLOQ), the value half of the LLOQ was used for calculations. Generally, mean values were calculated from the individually measured raw data. Fold changes with regard to the respective solvent control were determined and then log2-transformed (log2FC). A positive regulation is indicated by red shades, a negative regulation by blue shades. Statistical analysis was performed with Student’s *t*-test (*n* = 3, * *p* ≤ 0.05, ** *p* ≤ 0.01, *** *p* ≤ 0.001). Underlying fold changes and standard deviations can be found in the Appendix A “Underlying Data”.

**Figure 4 ijms-22-00372-f004:**
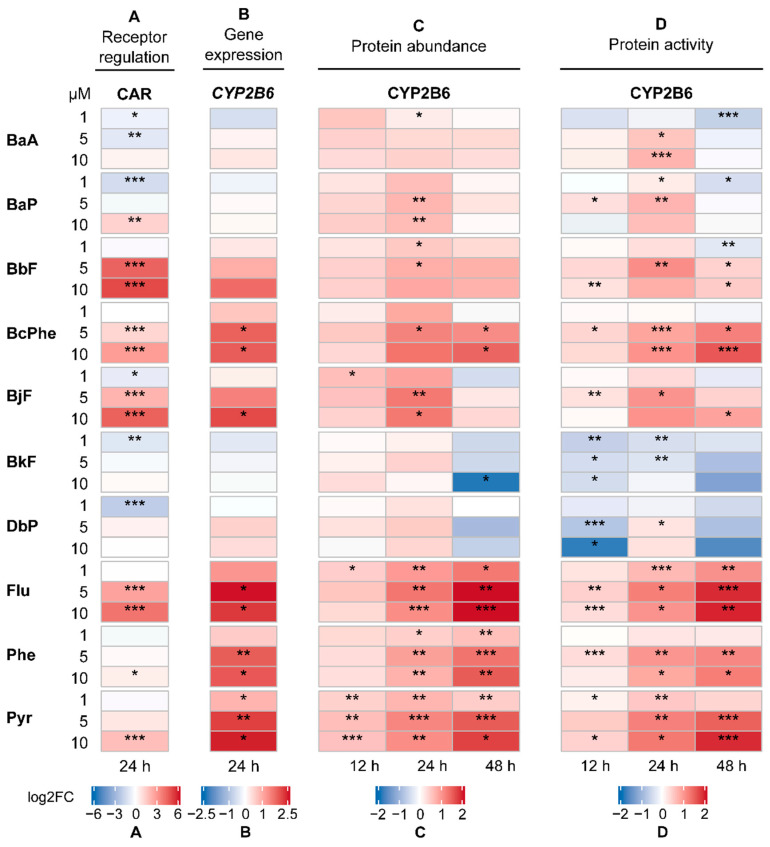
Cumulated heatmaps of receptor regulation (**A**), gene expression (**B**), protein abundance (**C**), and protein activity (**D**) of constitutive androstane receptor (CAR) and its target cytochrome P450 monooxygenase 2B6 (CYP2B6). Respective experiments were conducted as described in the Materials and Methods section. For protein abundance and activity, the following applied: For values below the respective lower limit of quantification (LLOQ), the value half of the LLOQ was used for calculations. Generally, mean values were calculated from the individually measured raw data. Fold changes with regard to the respective solvent control were determined and then log2-transformed (log2FC). A positive regulation is indicated by red shades, a negative regulation by blue shades. Statistical analysis was performed with Student’s *t*-test (*n* = 3, * *p* ≤ 0.05, ** *p* ≤ 0.01, *** *p* ≤ 0.001). Underlying fold changes and standard deviations can be found in in the Appendix A “Underlying Data”.

**Figure 5 ijms-22-00372-f005:**
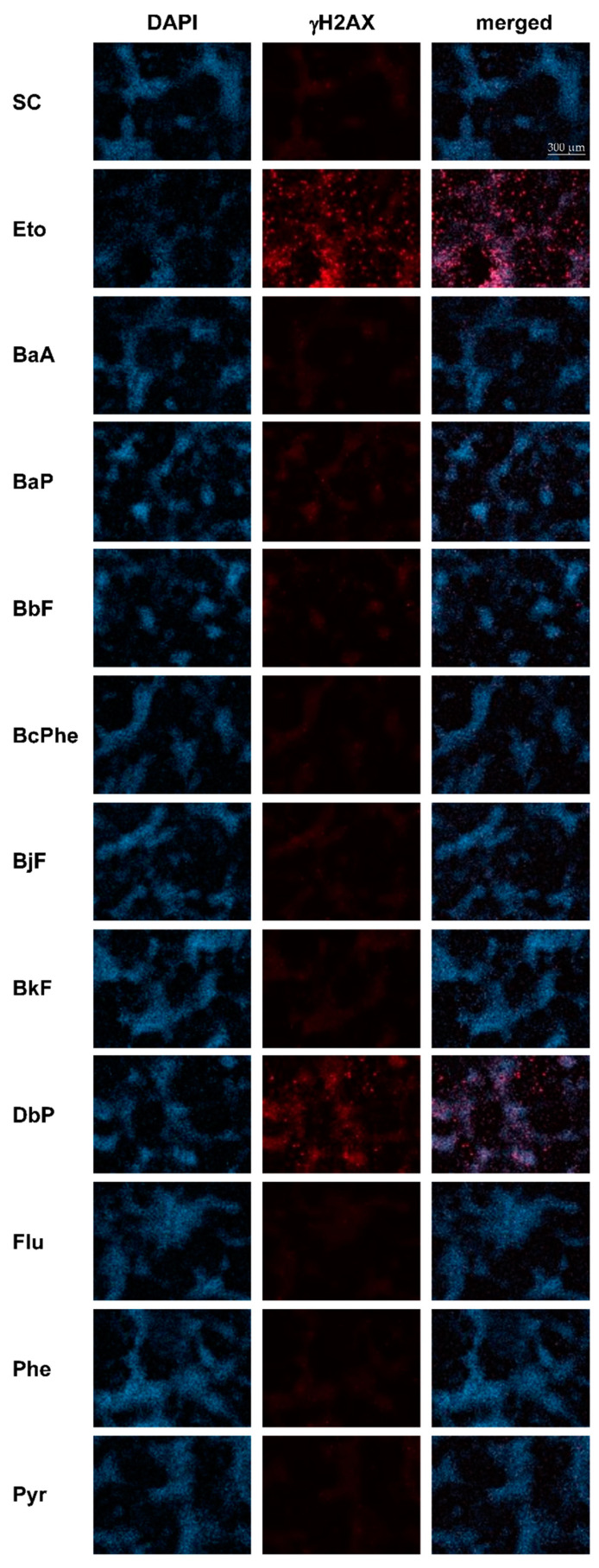
Frequency of histone H2AX phosphorylation. HepaRG cells were treated with 1, 5, or 10 µM of each PAH and incubated for 12, 24, 48 h. Etoposide (Eto; 25 µM) served as a metabolism-independent positive control. Afterward, cells were fixed, blocked, and stained with a combination of primary anti-γH2AX and secondary Alexa Fluor 647-antibody (red). Nuclei were counterstained using 4′,6-diamidino-2-phenylindole (DAPI) (blue). The figure depicts representative fluorescence images (brightness +30%) of each channel as well as a merged picture, respectively, 48 h posttreatment with solvent control (SC), Eto, or 10 µM PAH. Images were taken with Celldiscoverer (Zeiss, Oberkochen, Germany) and Zeiss Blue software (Zen 3.1 blue edition, Zeiss, Oberkochen, Germany).

**Figure 6 ijms-22-00372-f006:**
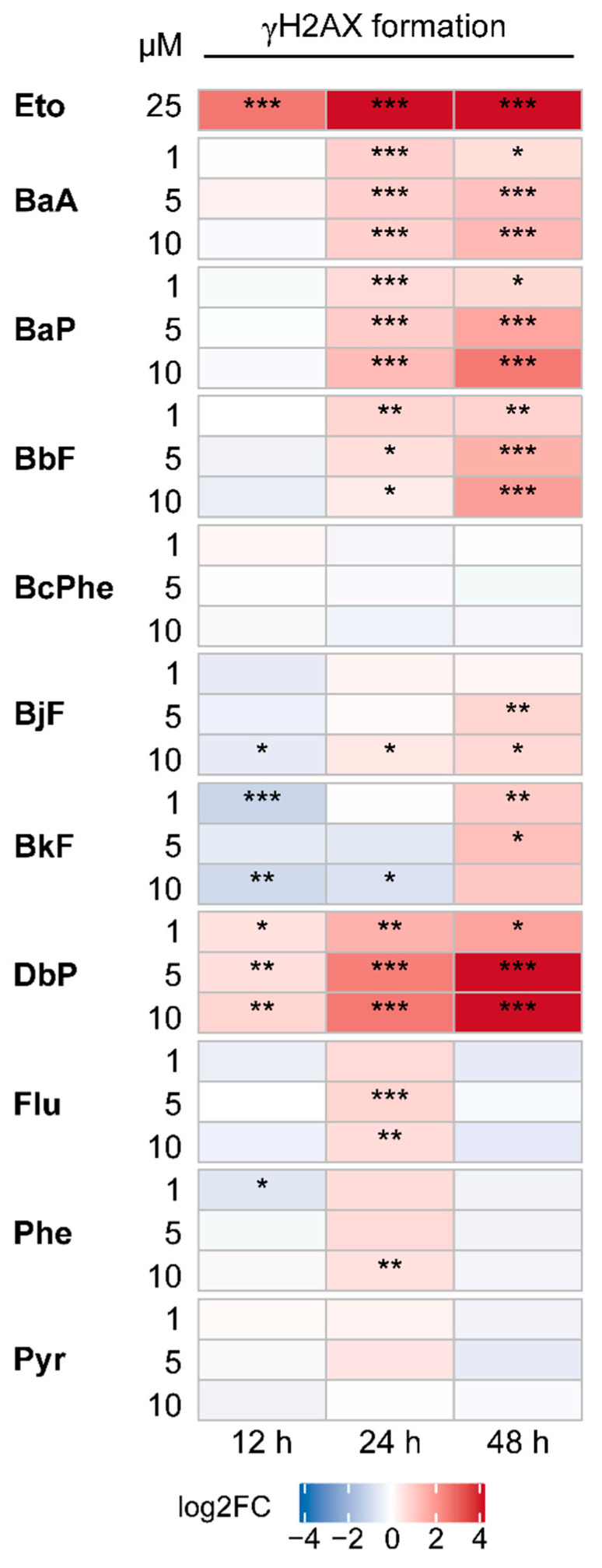
Quantitative analysis of H2AX phosphorylation presented as heatmap. Based on fluorescence images, fluorescence intensity was quantified. Following, fold induction over DAPI signal was calculated and referred to solvent control. Data were then log2_-_transformed (log2FC). An increase in H2AX phosphorylation is indicated by red shades, a decrease by blue shades. Statistical analysis was performed with Student’s *t*-test (*n* = 3, * *p* ≤ 0.05, ** *p* ≤ 0.01, *** *p* ≤ 0.001). Fold changes and standard deviations can be found in in the Appendix A “Underlying Data”.

## Data Availability

The data presented in this study are available in the Appendix A which can be found at https://www.mdpi.com/1422-0067/22/1/0/s1.

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
