# Peer review of "Polycyclic Aromatic Hydrocarbons Activate the Aryl Hydrocarbon Receptor and the Constitutive Androstane Receptor to Regulate Xenobiotic Metabolism in Human Liver Cells"

_ijms, 2020, doi:10.3390/ijms22010372_

Round 1
Reviewer 1 Report
General Comments: In this manuscript, the authors have analyzed relative potential of various polyaromatic hydrocarbon chemicals to induce AhR and CAR. Whereas the idea is interesting, the methodological flaws diminish the quality of the data. It is possible that at some dose each of these chemicals may have cross reactivity. Thus, the dose selection is critical and it is not clear how the used doses were selected. Further, confirmatory studies which include mutated luciferase construct as controls and cells without AhR and CAR are necessary.
Detailed Comments:
- Cytotoxicity data should be included in the main text.
- How were the doses decided? Are these toxicologically relevant doses?
- The abbreviations used for the chemicals should be included in Fig 1.
- The heatmaps are an interesting way of showing the data. However, it is hard to deduce quantitative data from this i.e. actual fold inductions. May be adding numbers to colors may help?
- What are the positive and negative controls for AhR and CAR activation? What are the colors in the heatmap refereeing to? Fold Change? Presumably! If yes, then what was used as the control to calculate them?
- The luciferase activity assay is doe in HepG2 cells, a hepatoblastoma cell line, which harbors a massive number of mutations. Some confirmation that these cells actually have WT AhR and CAR is necessary.
Reviewer 2 Report
This study demonstrates the effects of various PAHs to induce AHR and CAR mediated transcription and ultimately gene expression and enzymatic activity in two hepatocyte cell lines. The reviewer does have some points that need to be addressed:
- The data presentation, using heat maps to present all the data, is unconventional. There is a concern that some of the data is hidden within this presentation. While these R generated graphs are great for summation of data, please present the luciferase data, mRNA, protein level and protein activity as traditional bar graphs with data points and standard deviations presented. The individual data points are essential to see the spread of the data.
- Please validate the mass spec data for protein levels with western blotting, and present the western blots.
- γH2AX staining demonstrates the potential for apoptosis. Please validate these data with flow cytometry using Annexin V/PI staining.
- The data suggests that these compounds modulate AHR and CAR mediated transcription (by luciferase assay). Please show changes in AHR and CAR chromatin binding in response to compound exposure by ChIP.
- What are the effects of these compounds on primary hepatocytes? As primary hepatocytes can be cultured, the story would be strengthened to validate key findings in non-transformed cells.
Reviewer 3 Report
The authors aimed to investigate the activation of the receptors AHR and CAR by 10 polycyclic aromatic hydrocarbons (PAHs) in the human hepatic cells lines HepaRG and HepG2. They observed that overall, PAHs with a carcinogenic potential activated AHR as well as its targets, CYP1A, and were more likely to induce DNA damage. Non-carcinogenic PAHs primarily activated CAR and one of its target, CYP2B6.
The choice of HepaRG cells in this study is relevant since this cell line is metabolically competent, specially in terms of CYP activity.
The many results are respectably summarized and appropriately discussed.
All of these descriptive results are very interesting and will serve as a solid basis for future studies interested in HAP toxicity.
Round 2
Reviewer 1 Report
All comments have been satisfactorily addressed.
Reviewer 2 Report
While the authors responded in writing to all of the points that were initially raised, the authors failed to address the concerns and critiques.